# Simultaneous Analysis of Hydrophobic Atractylenolides, Atractylon and Hydrophilic Sugars in Bai-Zhu Using a High-Performance Liquid Chromatography Column Tandem Technique

**DOI:** 10.3390/foods12213931

**Published:** 2023-10-27

**Authors:** Zhixing Gu, Xi Nie, Ping Guo, Yuehua Lu, Bo Chen

**Affiliations:** Key Laboratory of Phytochemistry R&D of Hunan Province, Key Laboratory of Chemical Biology and Traditional Chinese Medicine Research of Ministry of Educational of China Institute of Interdisciplinary Studies, College of Chemistry and Chemical Engineering, Hunan Normal University, Changsha 410081, China; zhixingu@hunnu.edu.cn (Z.G.); 202120122346@hunnu.edu.cn (X.N.); guoping@hunnu.edu.cn (P.G.); lyh@hunnu.edu.cn (Y.L.)

**Keywords:** column tandem, reversed-phase liquid chromatography, hydrophilic interaction liquid chromatography, *Atractylodes macrocephala*, atractylenolides and atractylon, sugars

## Abstract

An analytical method was established using high-performance liquid chromatography coupled with diode array and evaporative light scattering detectors (HPLC-DAD-ELSD) with -C_18_ and -NH_2_ column tandem for the simultaneous determination of hydrophobic atractylenolide I, II, III, atractylone and hydrophilic compounds glucose, fructose and sucrose in the dried rhizome of *Atractylodes macrocephala* Koidz (a natural raw material for health foods, Bai-Zhu *aka*. in Chinese). The method combines the different separation capabilities of reversed-phase liquid chromatography and hydrophilic interaction liquid chromatography. It can provides a new choice for the simultaneous determination of hydrophilic and hydrophobic compounds in traditional Chinese medicines and health foods. It provided a reference method for the quality control of Bai-Zhu. The results showed that the linear correlation coefficients of the established column tandem chromatographic method were all greater than 0.9990, the relative standard deviation was 0.1–2.8%, and the average recovery was 96.7–103.1%. The contents of atractylenolide I, II, III, atractylone, fructose, glucose, and sucrose in 17 batches of Baizhu were 172.3–759.8 μg/g, 201.4–612.8 μg/g, 160.3–534.2 μg/g, 541.4–8723.1 μg/g, 6.9–89.7 mg/g, 0.7–7.9 mg/g, and 1.2–21.0 mg/g, respectively.

## 1. Introduction

*Atractylodes macrocephala* (Bai-Zhu *aka*. in Chinese) is the dried rhizome of *Atractylodes macrocephala* Koidz, which is widely used as a tonic herbal in eastern Asia. It belongs to the homologous resource of medicine and food in China [1,2,3,4]. The components include hydrophobic lactones, volatile oils, and hydrophilic carbohydrates, etc. Hydrophobic atractylenolide I, II, III and atractylon are important active components in Bai-Zhu, which have anti-tumor, anti-inflammation, anti-depression, anti-hypertension bio-activites [5,6]. The hydrophilic carbohydrate compounds in Bai-Zhu are also a kind of important component [7,8]. They are two kinds of important components in the quality control of Bai-Zhu.

In China, the unique processing (*aka*. Pao-Zhi in traditional Chinese medicine) of raw herbal medicines is a technique used to fulfill the different requirements of therapy, resulting in pharmacological properties changing, and toxicity or side effects reducing. There are obvious different in clinical efficacy between the raw and processed state [9]. Presently, the main processing methods of Bai-Zhu are steamed Bai-Zhu, stir-fried Bai-Zhu, soil-fried Bai-Zhu, soaked Bai-Zhu with rice swill, focal Bai-Zhu, bran-fried Bai-Zhu, and fried Bai-Zhu with honey bran [4]. Among the methods, bran-fried Bai-Zhu is documented in the Chinese Pharmacopeia [3]. Investigation on the changes occurring in the chemical components of Bai-Zhu before and after processing have been studied by utilizing HPLC, GC/MS, and other analytical techniques. Substantial changes in essential oil, as well as the sesquiterpenoids, e.g., atractylon, atractylnolides I, II, and III, etc., were observed. These changes can be explained as the result of evaporation, conversion, or degradation by heating, due to their physicochemical properties such as volatility and instability [10,11,12,13].

For the quality control of Bai-Zhu, as there is no quantitative identification of Bai-Zhu in the Chinese Pharmacopoeia [3]; research on the determination of the content of different chemical components in Bai-Zhu has been limited. Usually, analyses of atractylenolide I, II, III, atractylon and sugars in Bai-Zhu are completed using reversed-phase liquid chromatography (RP-HPLC) [14,15] and hydrophilic interaction liquid chromatography (HILIC) [16], independently [2]. The two independent methods are time-consuming and laborious. Therefore, it is important to develop a simple and rapid analytical method for the simultaneous detection of hydrophobic atractylenolide, atractylon, and hydrophilic carbohydrates in Bai-Zhu.

In order to simultaneously analyze hydrophilic and hydrophobic components in a complex matrix in the field of chromatography research, two-dimensional HPLC (2D-HPLC) methods and a column tandem technique have been employed based on the excellent orthogonality of RP-HPLC and HILIC [17,18,19,20,21,22,23,24,25,26,27]. Although the 2D-HPLC method allows for high separation resolution, the instrument system is highly complex. The column tandem technique can be employed on a simple instrument. The RP-LC column in tandem with an HILIC column has demonstrated that it is beneficial for the separation of hydrophilic and hydrophobic substances in complex samples, and has been successfully applied to the analysis of different targets in different sample matrices [20,21,22,23,24,25,26,27]. Therefore, the series of the RP-LC column and HILIC column, whose mobile phase (an acetonitrile-H_2_O system) is also compatible, are easy to achieve, and can be used for the simultaneous analysis of hydrophobic components atractylenolide I, II, III, atractylone and hydrophilic sugars in Bai-Zhu.

In this paper, an analytical method was established via HPLC-DAD-ELSD with a C_18_-NH_2_ column tandem for the simultaneous determination of hydrophobic components atractylenolide I, II, III, and atractylone, and hydrophilic compounds glucose, fructose and sucrose (Figure 1) in Bai-Zhu. The tandem of DAD and ELSD can simultaneously detect sugar with no UV absorption and atractylenolide I, II, III, and atractylone. It provides a reference method for the quality control of Bai-Zhu.

## 2. Materials and Methods

### 2.1. Materials and Instrumentation

The standards of atractylenolide I, II, III (purchased form Shanghai Yuanye Biotechnology Co., Shanghai, China) and atractylone (purchased form Sichuan Chengdu Aifa Biotechnology Co., Ltd., Chengdu, China) were >98% purity (HPLC). Fructose, glucose, and sucrose (>99%) were from Shanghai Maclean’s Biochemical Technology Co., Ltd. (Shanghai, China). All other chemicals are of analytical purity (purchased from Shanghai Aladdin Biochemical Technology Co.,Ltd., Shanghai, China). All medicinal slices of Bai-Zhu samples, including seven raw Bai-Zhu samples and ten processed Bai-Zhu (bran-fried Bai-Zhu) samples, were purchased in local pharmacies (Changsha, China).

Chromatographic analysis was performed with a Shimadzu 8040 HPLC system (Shimadzu, Kyoto, Japan) coupled to a SPD-M20A diode array detector (DAD) (Shimadzu, Kyoto, Japan) and an ELSD-LT II low-temperature evaporative light-scattering detector (ELSD) (Shimadzu, Kyoto, Japan). The separation was accomplished on an Inertsil^TM^ NH_2_ column (250 mm × 4.6 mm, 5 µm) (Shimadzu, Kyoto, Japan) and a Diamonsil C_18_ column (250 mm × 4.6 mm, 5 µm) (Dikma, Beijing, China).

A Thermo Scientific Sorvall ST 16R high-speed centrifuge (Thermo Fisher Scientific, Shanghai, China) was used for sample pretreatment. An F-030SD ultrasonic device (Shenzhen Fuyang Technology Group Co., Ltd., Shenzhen, China) was used for sample extraction.

### 2.2. Preparation of Standard Solutions

To construct the calibration curves, 0.50 mL, 1.0 mL, 4.0 mL, and 8.0 mL of a mixed standard solution with concentrations of 1.0 mg/mL, 1.0 mg/mL, 1.0 mg/mL, 0.025 mg/mL, 0.025 mg/mL, 0.025 mg/mL, 0.025 mg/mL for fructose, glucose, sucrose, atractylon, and atractylenolide I, II, and III were diluted with methanol to 10.0 mL. A series of mixed standard working solutions were obtained. Some 20 μL of the solution was injected into the HPLC, respectively. Linear regression was calculated using a least-squares method. All the above solutions were stored in a 4 °C refrigerator.

### 2.3. Sample Preparation

The medicinal slices of raw Bai-Zhu and processed Bai-Zhu were crushed into powder (20 mesh). Briefly, 0.20 g of the powder was extracted in 5 mL of methanol in an ultrasonic bath for 45 min. After centrifuging at 5000 rpm for 5 min, the supernatant was filtered using a 0.22 µm filter membrane. The filtrate was injected into HPLC. 

### 2.4. HPLC Conditions

The HPLC-DAD-ELSD conditions are as follows: an Inertsil^TM^ NH_2_ column (250 mm × 4.6 mm, 5 µm) was directly connected to a Diamonsil C_18_ column (250 mm × 4.6 mm, 5 µm) through a PEEK pipeline (30 mm in length). A gradient elution (solvent A: H_2_O, solvent B: acetonitrile (ACN)) was employed: 0–5 min 15% A, 5–12 min 15 to 20% A, 12–17 min 20 to 22% A, 17–32 min 22 to 30% A, 32–45 min 30 to 35% A, 45–65 min 35 to 40% A. The flow rate was 1 mL/min. UV detection wavelengths for quantitative analysis were set at 274 nm for atractylenolide I and 218 nm for atractylon, atractylenolide II, III, respectively. The ELSD conditions for sugar detection were as follows: the detector drift tube temperature was 50 °C, gas flow was 1.5 L/min, and signal gain was 8. The injection volume of sample solutions was 10 μL. 

## 3. Results and Discussion

### 3.1. Selection of Detectors

Alongside a big difference in polarity, there are also differences in spectral characteristics between sugars and terpenoids. It is well known that sugars do not have UV absorption, and cannot be analyzed using DAD. Because atractylon, and atractylenolide I, II, and III have high volatility and low content, they are difficult to detect through ELSD, because of their sensitivity. Therefore, the selection and combination of detectors is important for the simultaneous analysis of sugars and atractylon and atractylenolide I, II, and III in Bai-Zhu. 

For the quantitative analysis of atractylenolide I, II, III, and atractylenolide, the optimal UV detection wavelength was investigated. The UV absorption spectrum for each compound is shown in Figure 2. The characteristic absorption wavelength of atractylenolide I is 274 nm, and is the same for atractylenolide II, III and atractylon (218–219 nm). To obtain optimal sensitivity and anti-interference ability, 274 nm was selected as the optimal detection wavelength for the analysis of atractylenolide I, and 218 nm was selected as the optimal detection wavelength for atractylenolide II, III and atractylon in a number of reported methods [28,29,30,31]. The optimal selection of the detection wavelength can improve the sensitivity.

In this method, tandem DAD and ELSD was employed for the analysis of Bai-Zhu. As shown in Figure 3, the two kinds of compounds have very different responses in DAD and ELSD. It can be seen that both atractylenolide and atractylon have good responses on DAD, but there is no response on ELSD. On the contrary, sugars respond on ELSD and show no response on DAD. Therefore, the combination of DAD and ELSD is a good choice for the simultaneous detection of two kinds of compounds. In addition, the difference in responses according to the difference detector can also increase analytical selectivity due to incomplete separation.

### 3.2. Optimization of Separation Conditions

Using single-column mode for the separation, hydrophobic atractylenolide I, II, III, and atractylone can be separated on a C_18_ column based on hydrophobic interaction (a revised phase separation mode) (Figure 4a ELSD), and hydrophilic sugars can be separated on a -NH_2_ column based on hydrophilic interaction separation mode (Figure 4a PDA). The results were similar for Bai-Zhu sample solution (Figure 4b).

Sugars can not be separated completely on a -C_18_ column, and atractylenolide I, II, III and atractylone can not be separated completely on a NH_2_ column based on a H_2_O-ACN mobile phase system.

When a -C_18_ column is combined with a -NH_2_ column, two types of compounds can be separated completed. The chromatographic separation specification parameters, including retention factors (*k*), relative retention value (*α*), and resolutions ®, are as follows. *k*: atractylenolide III: 1.03, atractylenolide II: 1.32, atractylenolide I: 1.75, atractylone: 6.56, glucose: 3.10, fructose: 3.40, sucrose: 5.06. *α*: atractylenolide III: 1.00, atractylenolide II: 1.29, atractylenolide I: 1.71, atractylone: 6.39, glucose: 3.02, fructose: 3.31, sucrose: 4.93. R: atractylenolide III and atractylenolide II: 4.89, atractylenolide II and atractylenolide I: 4.82, glucose and fructose: 1.72. The results are shown in Figure 4.

In addition, two kinds of different connection, i.e., C_18_ columns in front of NH_2_ columns, and NH_2_ columns in front of C_18_ columns, on the simultaneous separation of atractylenolide I, atractylenolide II, atractylenolide III, atractylone, fructose, glucose, and sucrose were compared. The results show that there are no significant differences in separation efficiency and response intensity. 

### 3.3. Calibration Curves, LOD, and LOQ

Using the series of mixed standard working solutions, the linear regression of the method was completed. The LOD for seven compounds was 0.1–5.0 μg/mL (S/N = 3), and the LOQ was 1.25–50.0 μg/mL. The results are shown in Table 1.

### 3.4. Accuracy

Some 0.20 g of Bai-Zhu powder from the same batch with known contents was used, with six parallel portions, and a certain amount of reference solutions were added. The three spiked levels were 0.035, 0.35, and 0.625 mg/g for atractylenolide I, II, III, and atractylone in samples. The three spiked levels were 1.25, 12.5, 25.0 mg/g for glucose, fructose, and sucrose, respectively. Then, the extraction solution was obtained based on the preparation method described in Section 2.3, and analyzed using the HPLC method described in Section 2.4. The average recoveries of atractylenolide I, II, III, atractylone, glucose, fructose, and sucrose were in the range of 97.8–101.2%, 98.7–102.2%, 97.5–103.1%, 97.1–100.7%, 96.5–99.6%, 97.1–101.2%, and 96.7–100.1%, respectively. Additionally, all RSDs were within acceptable ranges. The results are shown in Table 2.

### 3.5. Stability

To investigate the sample solution stability, a same solution was analyzed at 0, 2, 4, 6, 8, 10, 12, and 24 h, respectively. The results show that RSDs of atractylenolide I, II, III, atractylone, glucose, fructose, and sucrose are 1.5%, 1.1%, 1.0%, 1.3%, 2.7%, 2.5% and 1.1%, respectively. This indicates that the extract of Bai-Zhu has good stability within 24 h.

### 3.6. Application

Briefly, 17 batches of Bai-Zhu sample were analyzed using the proposed C1_8_-NH_2_ column tandem HPLC-DAD-ELSD method. The results were also compared with those obtained using single-column methods such as RP-HPLC-DAD and HILIC-ELSD. The results are listed in Table 3.

From the results, it can be seen that (1) there are no significant differences between results obtained with the C_18_-NH_2_ column tandem HPLC-DAD-ELSD method and the RP-HPLC-DAD and HILIC-ELSD methods. (2) The unique processing of raw Bai-Zhu is one of the characteristics of traditional herbal medicines. Compared with raw materials, the content of atractylon in heat-prepared products decreased to different degrees (to almost ten times lower), and the contents of atractylenolide I, II, and III significantly increased. The reason for this is that atractylon is transformed under heating conditions, mainly into atractylenolides. Therefore, the heating process is very important for Bai-Zhu quality control [32,33,34]. However, there are obviously differences in the contents of the same types of materials. Because the raw and heat-processed Bai-Zhu were purchased from different local pharmacies at different times, it is possible that the brands of the samples and the harvesting season differed. This shows that standardization of plant materials is necessarily. (3) It has been found that the main low-molecular-weight sugar in Bai-Zhu is fructose. Moreover, oligosaccharides are abundant (Figure 4b). Therefore, the sugars in processed Bai-Zhu are more abundant than in raw materials due to the hydrolysis of oligosaccharides. In addition, the extract of Bai-Zhu has been shown to regulate disorders of glucose and lipid metabolism in mice with type 2 diabetes mellitus, enhancing insulin sensitivity, improving the level of inflammation, and alleviating liver injury and lipid deposition. The mechanism of these effects may be related to the up-regulation of the protein expression of GLP-1R [35]. Because glucose intake is very dangerous for people with type 2 diabetes, monitoring the sugars in Bai-Zhu is very important when Bai-Zhu is used for the treatment of type 2 diabetics.

## 4. Conclusions

In conclusion, a validated column tandem HPLC method was developed for the simultaneous determination of hydrophobic atractylenolide I, II, III, atractylone, and hydrophilic glucose, fructose and sucrose in Bai-Zhu, a natural raw material for health foods. The method combined two independent analytical procedures in one step. The results demonstrated that the proposed -C_18_ and -NH_2_ column tandem HPLC method had the same method specifications as independent analytical methods. Moreover, this method is simple, rapid, stable, and reliable for the routine monitoring of quality control components in Bai-Zhu. The method was successfully applied to real Bai-Zhu samples.

## Figures and Tables

**Figure 1 foods-12-03931-f001:**
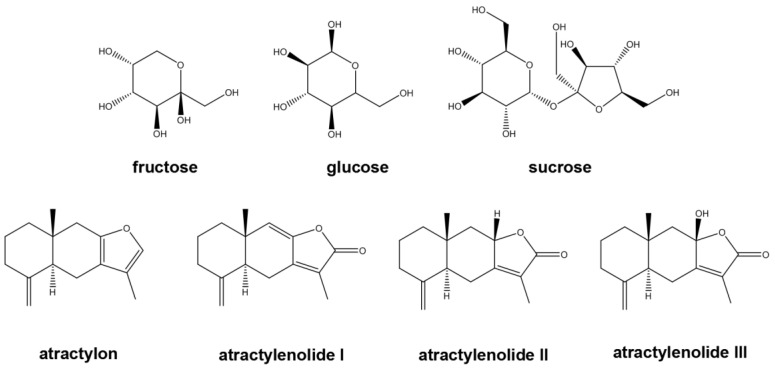
Chemical structure of the compounds.

**Figure 2 foods-12-03931-f002:**
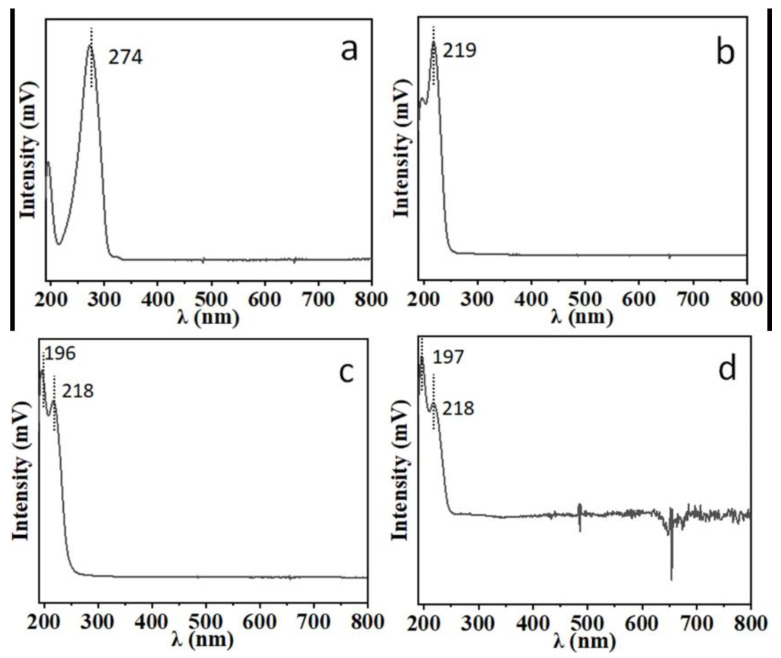
UV absorption spectrum of (**a**): atractylenolide I; (**b**): atractylenolide II; (**c**): atractylenolide III; (**d**): atractylon.

**Figure 3 foods-12-03931-f003:**
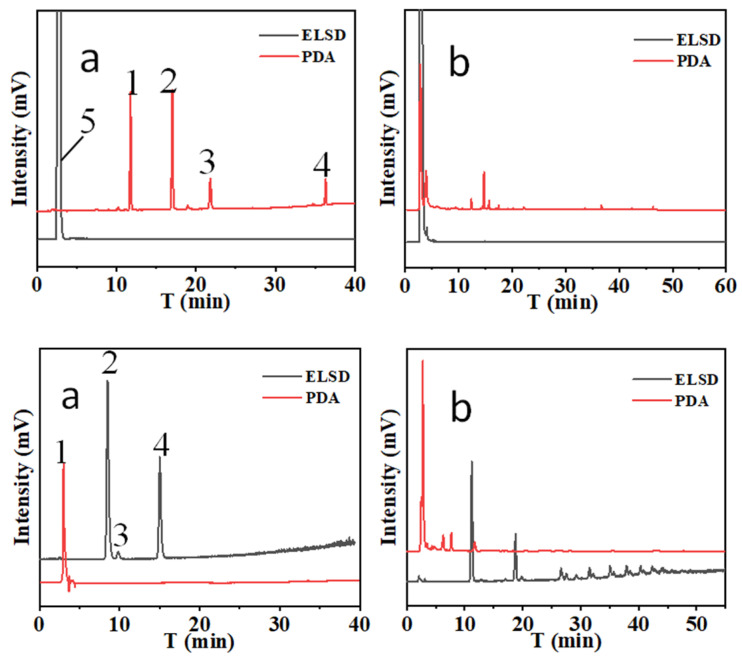
The chromatograms of atractylenolide I, II, III, atractylone and sugars on single column. (**upper**): -C_18_, 1: atractylenolide III, t_R_: 11.02 min.; 2: atractylenolide II, t_R_: 16.51 min.; 3: atractylenolide I, t_R_: 21.46 min.; 4: atractylone, t_R_: 36.78 min.; 5: sugars, t_R_: 2.13 min. (**lower**): -NH_2_, 1. terpenoids, t_R_: 2.51 min.; 2: fructose, t_R_: 8.32 min.; 3: glucose, t_R_: 9.45 min.; 4: sucrose, t_R_: 15.03 min). a: Mixed stanard solutions of atractylenolide I, II, III, atractylone and sugars; b: Bai-Zhu sample solution.

**Figure 4 foods-12-03931-f004:**
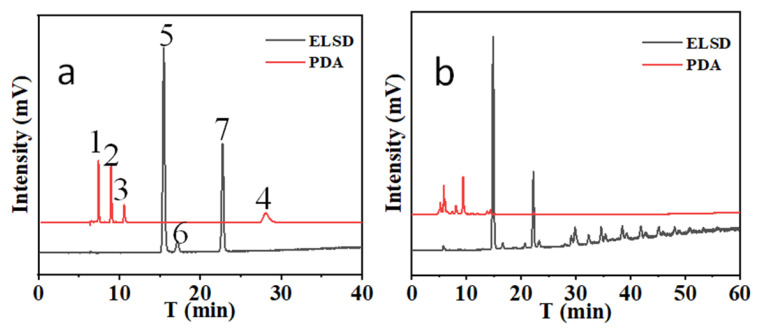
The chromatograms of mixed standard solution (**a**) and Bai-Zhu sample solution (**b**) on -C_18_ column in series with -NH_2_ column (1: atractylenolide III, t_R_: 7.54 min.; 2: atractylenolide II, t_R_: 8.64 min.; 3: atractylenolide I, t_R_: 10.24 min.; 4: atractylone, t_R_: 28.12 min.; 5: glucose, t_R_: 15.26 min.; 6: fructose, t_R_: 16.37 min.; 7: sucrose, t_R_: 22.56 min.).

**Table 1 foods-12-03931-t001:** Linearity, LODs and LOQs.

Analytes	Linear Range (μg/mL)	R^2^	LOD (μg/mL)	LOQ (μg/mL)	RSD (%)
Atractylenolide I	1.25–25.0	0.9999	0.1	1.25	0.12–0.3
Atractylenolide II	1.25–25.0	0.9997	0.1	1.25	0.3–1.4
Atractylenolide III	1.25–25.0	0.999	0.1	1.25	0.1–0.5
Atractylone	1.25–25.0	0.9999	0.1	1.25	0.2–1.1
Fructose	50–1000	0.9992	5	50	0.6–2.8
Glucose	50–1000	0.9996	5	50	0.3–2.6
Sucrose	50–1000	0.9992	5	50	0.7–1.7

**Table 2 foods-12-03931-t002:** Recovery of compounds.

Component	Spiked Level (mg/g)	Recovery Rate (%)	RSD (%)
Atractylenolide I	0.035	97.8 ± 2.0	2.0
	0.350	101.2 ± 1.3	1.3
	0.625	98.4 ± 1.7	1.7
Atractylenolide II	0.035	102.2 ± 1.1	1.1
	0.350	100.7 ± 1.9	1.9
	0.625	98.7 ± 2.1	2.1
Atractylenolide III	0.035	97.5 ± 1.5	1.5
	0.350	98.1 ± 2.3	2.3
	0.625	103.1 ± 2.4	2.3
Atractylone	0.035	97.1 ± 1.9	2.0
	0.350	100.7 ± 2.2	2.2
	0.625	99.2 ± 1.8	1.8
Fructose	1.25	98.4 ± 2.3	2.3
	12.5	101.2 ± 2.7	2.7
	25.0	97.1 ± 3.3	3.4
Glucose	1.25	96.5 ± 2.7	2.8
	12.5	97.6 ± 2.9	3.0
	25.0	99.6 ± 3.5	3.5
Sucrose	1.25	99.3 ± 2.1	2.1
	12.5	100.1 ± 2.5	2.5
	25.0	96.7 ± 2.4	2.5

**Table 3 foods-12-03931-t003:** The determination results of Bai-Zhu samples (*n* = 3).

Sample	A-I (μg/g) (t-H/H-D)	A-II (μg/g) (t-H/H-D)	A-III (μg/g) (t-H/H-D)	A (μg/g) (t-H/H-D)	Fru (mg/g) (t-H/H-E)	Glu (mg/g) (t-H/H-E)	Suc (mg/g) (t-H/H-E)
H-P 1	460.0 ± 5.1/ 462.7 ± 4.9	412.3 ± 4.2/ 407.2 ± 4.6	380.6 ± 3.4/ 389.5 ± 3.8	642.7 ± 6.3/ 651.6 ± 5.2	44.3 ± 1.1/ 45.1 ± 0.9	2.6 ± 0.1/ 2.5 ± 0.1	17.5 ± 1.1/ 17.4 ± 1.1
H-P 2	531.5 ± 7.2/ 534.7 ± 6.4	487.9 ± 3.7/ 480.5 ± 4.1	423.1 ± 5.1/ 431.0 ± 3.9	553.3 ± 6.2/ 549.7 ± 5.2	26.3 ± 0.2/ 26.1 ± 0.3	1.8 ± 0.1/ 1.8 ± 0.1	19.1 ± 0.5/ 19.0 ± 0.4
H-P 3	713.3 ± 9.4/ 721.4 ± 10.0	562.4 ± 7.2/ 571.8 ± 6.5	534.2 ± 4.3/ 545.3 ± 5.5	579.4 ± 7.7/ 582.7 ± 6.9	23.0 ± 0.4 23.5/ ± 0.5	1.8 ± 0.1/ 1.9 ± 0.1	11.4 ± 0.2/ 11.2 ± 0.1
H-P 4	504.6 ± 10.2/ 512.8 ± 9.2	447.8 ± 8.9/ 462.3 ± 9.2	423.9 ± 7.3/ 432.6 ± 6.1	611.2 ± 11.4/ 630.4 ± 12.3	14.1 ± 0.4/ 13.9 ± 0.3	0.9 ± 0.1/ 0.8 ± 0.1	5.3 ± 0.2/ 5.6 ± 0.2
H-P 5	641.2 ± 14.3/ 652.7 ± 12.1	487.3 ± 9.8/ 477.6 ± 10.2	312.4 ± 6.1/ 322.5 ± 7.1	621.4 ± 11.7/ 612.5 ± 10.4	46.3 ± 2.0/ 45.2 ± 2.1	3.1 ± 0.2/ 3.0 ± 0.1	21.0 ± 0.5/ 21.6 ± 0.6
H-P 6	552.0 ± 10.9/ 560.3 ± 8.7	510.3 ± 9.3/ 511.8 ± 7.2	462.7 ± 8.2/ 470.4 ± 9.2	567.7 ± 6.5/ 570.2 ± 7.1	13.4 ± 0.5/ 13.0 ± 0.5	1.2 ± 0.1/ 1.1 ± 0.1	3.2 ± 0.1/ 3.2 ± 0.1
H-P 7	520.4 ± 13.4/ 523.2 ± 11.3	410.6 ± 10.3/ 413.7 ± 8.2	365.2 ± 11.5/ 367.2 ± 7.3	773.6 ± 21.4/ 768.1 ± 18.2	13.1 ± 0.3/ 13.1 ± 0.4	0.7 ± 0.1/ 0.6 ± 0.0	5.1 ± 0.1/ 5.2 ± 0.1
H-P 8	659.7 ± 13.2/ 666.2 ± 14.2	602.6 ± 21.2/ 610.9 ± 18.7	421.6 ± 11.0/ 425.8 ± 8.7	666.2 ± 17.8/ 657.1 ± 19.2	16.0 ± 0.3/ 16.2 ± 0.2	1.3 ± 0.0/ 1.3 ± 0.0	9.0 ± 0.2/ 8.7 ± 0.3
H-P 9	606.5 ± 23.7/ 610.2 ± 22.1	432.7 ± 16.2/ 440.2 ± 17.1	379.8 ± 7.2/ 382.5 ± 7.0	634.8 ± 16.2/ 641.4 ± 15.2	89.7 ± 2.3/ 88.7 ± 3.0	7.9 ± 0.2/ 7.8 ± 0.2	7.4 ± 0.1/ 7.4 ± 0.1
H-P 10	759.8 ± 28.9/ 756.2 ± 26.3	612.8 ± 22.1/ 620.1 ± 18.2	489.4 ± 18.9/ 488.0 ± 15.2	541.4 ± 10.4/ 548.3 ± 8.5	21.0 ± 0.6/ 21.8 ± 0.5	1.9 ± 0.1/ 1.9 ± 0.1	2.7 ± 0.1/ 2.6 ± 0.1
Raw 1	172.3 ± 5.3/ 175.2 ± 4.2	181.8 ± 3.2/ 177.9 ± 4.1	160.3 ± 5.7/ 162.3 ± 4.6	7682.2 ± 230.6/ 7699.1 ± 210.7	11.2 ± 0.3/ 11.4 ± 0.4	1.0 ± 0.0/ 1.1 ± 0.0	2.9 ± 0.1/ 2.9 ± 0.1
Raw 2	203.0 ± 6.9/ 209.4 ± 6.1	214.7 ± 7.2/ 216.3 ± 6.4	210.8 ± 6.0/ 211.6 ± 5.2	6871.2 ± 210.1/ 6789.3 ± 220.3	8.7 ± 0.2/ 8.8 ± 0.2	0.7 ± 0.0/ 0.7 ± 0.0	1.4 ± 0.0/ 1.3 ± 0.0
Raw 3	274.1 ± 5.3/ 268.2 ± 6.4	252.6 ± 7.5/ 257.3 ± 5.2	237.6 ± 8.1/ 234.2 ± 6.3	5679.7 ± 200.1/ 5690.4 ± 189.5	12.1 ± 0.4/ 12.3 ± 0.3	1.2 ± 0.0/ 1.2 ± 0.0	2.8 ± 0.0/ 2.9 ± 0.0
Raw 4	216.5 ± 5.0/ 220.7 ± 5.3	301.3 ± 4.2/ 305.4 ± 5.5	243.3 ± 7.0/ 240.8 ± 4.1	5995.1 ± 203.6/ 6004.7 ± 178.3	6.9 ± 0.1/ 6.8 ± 0.1	1.0 ± 0.0/ 1.0 ± 0.0	1.2 ± 0.0/ 1.2 ± 0.0
Raw 5	185.8 ± 7.0/ 180.1 ± 6.8	224.9 ± 8.3/ 230.1 ± 6.8	201.7 ± 7.7/ 200.0 ± 7.1	8723.1 ± 236.6/ 8834.9 ± 229.5	23.1 ± 0.7/ 23.4 ± 0.5	1.6 ± 0.0/ 1.5 ± 0.0	2.3 ± 0.0/ 2.1 ± 0.0
Raw 6	217.0 ± 4.1/ 220.4 ± 5.0	201.4 ± 6.7/ 206.5 ± 7.1	235.7 ± 4.3/ 240.1 ± 5.6	7213.6 ± 172.0/ 7309.8 ± 183.7	11.7 ± 0.4/ 12.2 ± 0.3	0.8 ± 0.0/ 0.8 ± 0.0	1.5 ± 0.0/ 1.5 ± 0.0
Raw 7	249.7 ± 3.7/ 253.6 ± 4.9	222.6 ± 8.5/ 228.4 ± 7.4	201.4 ± 5.5/ 196.3 ± 6.2	6102.5 ± 217.1/ 6185.4 ± 234.5	9.2 ± 0.3/ 9.3 ± 0.3	1.2 ± 0.0/ 1.2 ± 0.0	2.6 ± 0.0/ 2.5 ± 0.0

Note: A-I: Atractylenolide I; A-II: Atractylenolide II; A-III: Atractylenolide III; A: Atractylone; Fru: Fructose; Glu: Glucose; Suc: Sucrose; t-H/H-D: tandem HPLC/HPLC-DAD; t-H/H-E: tandem HPLC/HILIC-ELSD; H-P: Heat-processed.

## Data Availability

Data is contained within the article.

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
