# Peer review of "Simultaneous Analysis of Hydrophobic Atractylenolides, Atractylon and Hydrophilic Sugars in Bai-Zhu Using a High-Performance Liquid Chromatography Column Tandem Technique"

_foods, 2023, doi:10.3390/foods12213931_

Round 1

Reviewer 1 Report

Comments and Suggestions for Authors

Review of the manuscript entitled: “Simultaneous analysis of hydrophobic atractylenolides, atractylon and hydrophilic sugars in Bai-Zhu by HPLC column tandem technique.

The manuscript presents the procedure of simultaneous analysis of two kind of compounds using the tandem HPLC columns with two modes of the detection. The separation of two various classes compounds namely hydrophobic (atractylenoides) and hydrophilic (sugars) in one procedure is very difficult due to various physicochemical properties of investigated solutes. Basing on the literature research the authors proposed two modes of separation: reversed phase (C-18 column) and HILIC (NH2) combined into one HPLC system and with two modes of the detection: (spectrophotometric at UV range for atractylenoides and evaporative light scattering for sugars). The presented experiments convicted reader the reason of choosing both column and detector types. The method was partially validated (the linear range of the determination, LOD and LOQ, accuracy and stability) and then it was applied to determine the investigated compounds in the real samples (Bai-Zhu; dried rhizome of Atractylodes macrocephala Koidz).   The manuscript is well prepared however, it has some data to be improved or explained and included.

The following sentences are not clear for the reader and should be re-wrote:

In Abstract part: “The method combines different separation capabilities of reversed--phase liquid chromatography and hydrophilic interaction liquid chromatography, can provide a new choice for the simultaneous determination of hydrophilic and hydrophobic compounds in traditional Chinese medicines and health foods”. Reviewer suggests divide it into two sentences.

In Introduction part: “Among the hydrophobic components of atractylodes, the contents of atractylenolide I, II, III and atractylon are relatively high, which is an important active component of Bai-Zhu, and plays an important role in anti-tumor, anti-inflammation, anti-depression, anti-hypertension etc”. This sentence is not clear.

“Therefore, it is significant to develop a simple and rapid analytical method for simultaneous detection of hydrophobic atractylenolide, atractylon, and hydrophilic carbohydrates in Bai-Zhu for its quality control”. This sentence should be divided into two.

“The RP-LC column in tandem with HILIC column is beneficial to the separation of polar and non-polar substances in the mixture samples, and has been successfully applied to the analysis of phenols in wine [17], mouse serum metabolites [18], drugs [19] etc.”

It is not clear the manuscript authors idea for the ref. 18 content. According to the presented text the paper from ref. 18 determine metabolites of mouse serum. The reviewer checked the paper and it investigated metabolites in mouse serum.  

In the Materials and Instrumentation part:

“Shimadzu 8040 HPLC system (Shimadzu, Japan) was employed which containing SPD-M20A

Diode array detector (DAD) and ELSD-LT II low temperature-evaporative light scattering

Detector (ELSD)”. The sentence is not clear.

Result and discussion part:

“ Characteristic absorption wavelength of atractylenolide I is 274 nm, one of atractylenolide II, III and atractylon is 218 -219 nm”. It should be:  “Characteristic absorption wavelength of atractylenolide I is 274 nm, and the same for atractylenolide II, III and atractylon (218 - 219 nm)”.

“In addition, to investigating whether the connection order of chromatographic columns in column series affect the separation and response of the compounds, two modes of C18 columns in front of NH2 columns and NH2 columns in front of C18 columns on the simultaneous separation of atractylenolide I, atractylenolide II, atractylenolide III, atractylone, fructose, glucose, and sucrose were tested”. This sentence is not clear.

Part: Calibration Curves, LOD, and LOQ

“The LOD for seven compounds was 0.1 - 5.0 μg/mL (S/N=3), and the LOQ was 1.20 - 50.0 μg/mL. The results are shown in Table 1”…

As for the LOQ the value presented in Table 1 was 1.25 µg/mL not 1.20 µg/mL.

Part: Application

The Table 3 content is not fully explained in the text. The reader does not know what are Heat processing procedures 1-10 and raw 1-7. The full explanation  for these data  should be included into the manuscript. It looks like some experiments were not presented in the text.

 There is no cover letter which according to the instruction for the authors should be included.

 Table 3 does not contain data corresponding to the following text: “ From the results, it can be seen that: (1) there are no significant differences on results between that obtained from C18--NH2 column tandem HPLC--DAD--ELSD and RP--HPLC--DAD (the results are listed) and HILIC--ELSD (the results are listed)”

 The sentence: Chinese medicine must be processed before it can be used as medicine, which is one of the characteristics of Chinese medicine” … is not clear for the reader.

 There are some grammar mistakes in the text.

It is weird for the reviewer to find the article drift before its reviewing in the preprints.org.   (see doi: 10.20944/preprints202309.1011.v1).

 According to the reviewer the presented manuscript should be significantly improved. In the present form cannot be recommended.  

Comments on the Quality of English Language

There are some grammar mistakes in the text which should be improved.

Author Response

Revewer 1

Review of the manuscript entitled: “Simultaneous analysis of hydrophobic atractylenolides, atractylon and hydrophilic sugars in Bai-Zhu by HPLC column tandem technique.

The manuscript presents the procedure of simultaneous analysis of two kind of compounds using the tandem HPLC columns with two modes of the detection. The separation of two various classes compounds namely hydrophobic (atractylenoides) and hydrophilic (sugars) in one procedure is very difficult due to various physicochemical properties of investigated solutes. Basing on the literature research the authors proposed two modes of separation: reversed phase (C-18 column) and HILIC (NH2) combined into one HPLC system and with two modes of the detection: (spectrophotometric at UV range for atractylenoides and evaporative light scattering for sugars). The presented experiments convicted reader the reason of choosing both column and detector types. The method was partially validated (the linear range of the determination, LOD and LOQ, accuracy and stability) and then it was applied to determine the investigated compounds in the real samples (Bai-Zhu; dried rhizome of Atractylodes macrocephala Koidz). The manuscript is well prepared however, it has some data to be improved or explained and included.

The following sentences are not clear for the reader and should be re-wrote:

In Abstract part: “The method combines different separation capabilities of reversed--phase liquid chromatography and hydrophilic interaction liquid chromatography, can provide a new choice for the simultaneous determination of hydrophilic and hydrophobic compounds in traditional Chinese medicines and health foods”. Reviewer suggests divide it into two sentences.

Answer: It has been revised. Thanks.

In Introduction part: “Among the hydrophobic components of atractylodes, the contents of atractylenolide I, II, III and atractylon are relatively high, which is an important active component of Bai-Zhu, and plays an important role in anti-tumor, anti-inflammation, anti-depression, anti-hypertension etc”. This sentence is not clear.

Answer: It has been revised. Thanks.

“Therefore, it is significant to develop a simple and rapid analytical method for simultaneous detection of hydrophobic atractylenolide, atractylon, and hydrophilic carbohydrates in Bai-Zhu for its quality control”. This sentence should be divided into two.

Answer: It has been revised. Thanks.

“The RP-LC column in tandem with HILIC column is beneficial to the separation of polar and non-polar substances in the mixture samples, and has been successfully applied to the analysis of phenols in wine [17], mouse serum metabolites [18], drugs [19] etc.”It is not clear the manuscript authors idea for the ref. 18 content. According to the presented text the paper from ref. 18 determine metabolites of mouse serum. The reviewer checked the paper and it investigated metabolites in mouse serum.

Answer: Ahthors just described column tandem technique have been applied different areas successfully. The paragraph has been revised.. Thanks.

In the Materials and Instrumentation part:

“Shimadzu 8040 HPLC system (Shimadzu, Japan) was employed which containing SPD-M20A Diode array detector (DAD) and ELSD-LT II low temperature-evaporative light scattering Detector (ELSD)”. The sentence is not clear.

Answer: It has been revised. Thanks.

Result and discussion part:

“ Characteristic absorption wavelength of atractylenolide I is 274 nm, one of atractylenolide II, III and atractylon is 218 -219 nm”. It should be: “Characteristic absorption wavelength of atractylenolide I is 274 nm, and the same for atractylenolide II, III and atractylon (218 - 219 nm)”.

Answer: It has been revised. Thanks.

“In addition, to investigating whether the connection order of chromatographic columns in column series affect the separation and response of the compounds, two modes of C18 columns in front of NH2 columns and NH2 columns in front of C18 columns on the simultaneous separation of atractylenolide I, atractylenolide II, atractylenolide III, atractylone, fructose, glucose, and sucrose were tested”. This sentence is not clear.

Answer: It has been revised. Thanks.

Part: Calibration Curves, LOD, and LOQ

“The LOD for seven compounds was 0.1 - 5.0 μg/mL (S/N=3), and the LOQ was 1.20 - 50.0 μg/mL. The results are shown in Table 1”…

As for the LOQ the value presented in Table 1 was 1.25 µg/mL not 1.20 µg/mL.

Answer: It has been revised. Thanks.

Part: Application

The Table 3 content is not fully explained in the text. The reader does not know what are Heat processing procedures 1-10 and raw 1-7. The full explanation for these data should be included into the manuscript. It looks like some experiments were not presented in the text.

Answer: It has been revised. Thanks.

There is no cover letter which according to the instruction for the authors should be

included.

Answer: We put the cover letter on the website when submitted the manuscript. Thanks.

Table 3 does not contain data corresponding to the following text: “ From the results, it can be seen that: (1) there are no significant differences on results between that obtained from C18--NH2 column tandem HPLC--DAD--ELSD and RP--HPLC--DAD (the results are listed) and HILIC--ELSD (the results are listed)”

Answer: We are very sorry. It’s our negligence. The data have been added in Table 3. Pls check it. Thanks.

The sentence: Chinese medicine must be processed before it can be used as medicine, which is one of the characteristics of Chinese medicine” … is not clear for the reader.

Answer: It has been revised. In revised introduction section, we has added some content to explain the processing of raw materials in traditional Chinese medicine. Thanks.

There are some grammar mistakes in the text.

Answer: We have polished the manuscript. Pls check the reversion version. Thanks.

It is weird for the reviewer to find the article drift before its reviewing in the preprints.org. (see doi: 10.20944/preprints202309.1011.v1).

Answer: After submitting the draft to Foods, we received a email about the preprint from Preprints Editorial Office at https://www.preprints.org/. We were also confused, and agreed it. So sorry. Pls check the Screenshot of email. Thanks.

According to the reviewer the presented manuscript should be significantly improved. In the present form cannot be recommended.

Answer: We have modified the manuscript. Pls check the reversion version. Thank you so much for your rigorous comments.

Reviewer 2 Report

Comments and Suggestions for Authors

Observations!

  1. Paragraph: „To construct the calibration curves, 0.50 mL, 1.0 mL, 4.0 mL, and 8.0 mL of a mixed standard solution with concentrations of 1.0 mg/mL, 1.0 mg/mL, 1.0 mg/mL, 0.025 mg/mL,.......” , from chapter 3.3., must be reformulated, because it was described in chapter 2.2.( Preparation of standard solutions).
  1. In chapter 2.3 "Sample Preparation", the types of analyzed samples must be presented: (processed and raw, table 3).
  1. Do the results in table 3 show average values? It must be presented as mean±standard deviation.

Author Response

Revewer 2

Observations!

Paragraph: „To construct the calibration curves, 0.50 mL, 1.0 mL, 4.0 mL, and 8.0 mL of a mixed standard solution with concentrations of 1.0 mg/mL, 1.0 mg/mL, 1.0 mg/mL, 0.025 mg/mL,.......” , from chapter 3.3., must be reformulated, because it was described in chapter 2.2.( Preparation of standard solutions).

Answer: It has been revised. Thanks.

In chapter 2.3 "Sample Preparation", the types of analyzed samples must be presented: (processed and raw, table 3).

Answer: It has been revised. Thanks.

Do the results in table 3 show average values? It must be presented as mean±standard deviation.

Answer: It has been revised. Thanks.

Reviewer 3 Report

Comments and Suggestions for Authors

This manuscript describes the development of an analytical method using HPLC-DAD-ELSD with a C18-NH2 column tandem for the simultaneous quantification of both hydrophobic and hydrophilic compounds in the dried rhizome of Atractylodes macrocephala Koidz (commonly known as Bai-Zhu in Chinese), which is a natural raw material used in health foods.

The method combines the separation capabilities of reversed-phase liquid chromatography and hydrophilic interaction liquid chromatography, offering a novel approach for assessing the content of these compounds in traditional Chinese medicines and health foods. This method serves as a valuable reference for quality control of Bai-Zhu. The results demonstrate high accuracy and precision, with linear correlation coefficients above 0.9990, relative standard deviations ranging from 0.1% to 2.8%, and average recoveries between 96.7% and 103.1%. The content of various compounds, including atractylenolides I, II, III, atractylone, fructose, glucose, and sucrose, in 17 batches of Bai-Zhu samples was also determined and found to vary within specified ranges.

I suggest to the authors to check their manuscript and refine the language carefully, if necessary, with a native speaker, as the level of English throughout the manuscript is scarce. There are a number of grammatical errors and instances of badly worded/constructed sentences. Besides, some parts are really confused, and the text is not always comprehensible. This makes the reader confused and makes this work uninteresting.

Author Response

This manuscript describes the development of an analytical method using HPLC-DAD-ELSD with a C18-NH2 column tandem for the simultaneous quantification of both hydrophobic and hydrophilic compounds in the dried rhizome of Atractylodes macrocephala Koidz (commonly known as Bai-Zhu in Chinese), which is a natural raw material used in health foods.

The method combines the separation capabilities of reversed-phase liquid chromatography and hydrophilic interaction liquid chromatography, offering a novel approach for assessing the content of these compounds in traditional Chinese medicines and health foods. This method serves as a valuable reference for quality control of Bai-Zhu. The results demonstrate high accuracy and precision, with linear correlation coefficients above 0.9990, relative standard deviations ranging from 0.1% to 2.8%, and average recoveries between 96.7% and 103.1%. The content of various compounds, including atractylenolides I, II, III, atractylone, fructose, glucose, and sucrose, in 17 batches of Bai-Zhu samples was also determined and found to vary within specified ranges.

I suggest to the authors to check their manuscript and refine the language carefully, if necessary, with a native speaker, as the level of English throughout the manuscript is scarce. There are a number of grammatical errors and instances of badly worded/constructed sentences. Besides, some parts are really confused, and the text is not always comprehensible. This makes the reader confused and makes this work uninteresting.

Answer: Thanks for your constructive suggestion. We have revised and polished the manuscript. Pls check the revised version.

Reviewer 4 Report

Comments and Suggestions for Authors

This paper describes a simultaneous analysis of hydrophobic atractylenolides, atractylon and hydrophilic sugars in Bai-Zhu by HPLC column tandem technique. The article is quite complete, it is of interest to the scientific community, the methods and statistics used are appropriate and the results and discussion are conveniently described. The English language is correct. The work is interesting and delves in the simultaneous separation of compounds from Bai-Zhu.  

I consider that the article is appropriate to be recommended in Foods journal once the authors have made some modifications to it.

Major aspects:

-          The authors must include a study on the characteristics of the chromatographic separation of the peaks (retention time, resolution, k, α, etc.).

-          Include a conclusion section 

Minor aspects: 

-          Authors must follow the journal's format.

-          Define DAD-ELSD the first time.

-          Materials and methods: Include the city and country of all the companies cited, and cite the companies of all the reagents and equipment’s employed. In case of USA companies, include the city and the state abbreviation. Unify and apply to the entire document. 

Comments on the Quality of English Language

Minor editing of English language required

Author Response

Revewer 4

This paper describes a simultaneous analysis of hydrophobic atractylenolides, atractylon and hydrophilic sugars in Bai-Zhu by HPLC column tandem technique. The article is quite complete, it is of interest to the scientific community, the methods and statistics used are appropriate and the results and discussion are conveniently described. The English language is correct. The work is interesting and delves in the simultaneous separation of compounds from Bai-Zhu.

I consider that the article is appropriate to be recommended in Foods journal once the authors have made some modifications to it.

Major aspects:

The authors must include a study on the characteristics of the chromatographic separation of the peaks (retention time, resolution, k, α, etc.).

Answer: They have been added. Thanks.

Include a conclusion section

 Answer: It has been added. Thanks.

Minor aspects:

Authors must follow the journal's format.

Answer: The submission requirement mentions that the manuscript can be submitted in normal word format, not only in journal template. Thanks.

Define DAD-ELSD the first time.

Answer: It has been defined in the revised abstract. Pls check it. Thanks.

Materials and methods: Include the city and country of all the companies cited, and cite the companies of all the reagents and equipment’s employed. In case of USA companies, include the city and the state abbreviation. Unify and apply to the entire document.

Answer: They have been added. Pls check it. Thanks.

Reviewer 5 Report

Comments and Suggestions for Authors

Manuscript by Zhixing at al. concerns the validation of the method for the separation of polar and hydrophobic compounds using a tandem system of columns and detectors. It is quite an original work, but the entire manuscript is written quite poorly and cannot be recommended at this stage.

First of all, the introduction does not provide a good background on the topic. The authors focused more on citing various examples of the use of HPLC in a 2D system than on the characteristics of the raw material itself and the significance of the research undertaken in terms of its impact on current science. In fact, the introduction, starting from the second paragraph, is more open to discussion.

The methodological part is written correctly, but there is no detailed description of the validation procedure used. Figure 1 should be included either in the introduction or in the discussion of results section. There are no other details, such as the characteristics of the centrifuge or ultrasonic divace.

Chapter 3 Results and Discussion is basically just a description of the results. Discussion of other authors' results is virtually non-existent in this chapter. The last two sentences are more suitable for the introduction.

No conclusions from the research.

Very poor bibliography.

When discussing the results, a full characterization of the peaks is missing, including retention times.

Author Response

Revewer 5

Manuscript by Zhixing at al. concerns the validation of the method for the separation of polar and hydrophobic compounds using a tandem system of columns and detectors. It is quite an original work, but the entire manuscript is written quite poorly and cannot be recommended at this stage.

Answer: Thank you so much for your rigorous comments. We have revised the manuscript. Pls check the reversion version.

First of all, the introduction does not provide a good background on the topic. The authors focused more on citing various examples of the use of HPLC in a 2D system than on the characteristics of the raw material itself and the significance of the research undertaken in terms of its impact on current science. In fact, the introduction, starting from the second paragraph, is more open to discussion.

Answer: We have modified the introduction. Pls check the reversion version. Thank you so much for your rigorous comments.

The methodological part is written correctly, but there is no detailed description of the validation procedure used. Figure 1 should be included either in the introduction or in the discussion of results section. There are no other details, such as the characteristics of the centrifuge or ultrasonic divace.

Answer: Figure 1 has been moved in introduction. Other information has been added. Thanks.

Chapter 3 Results and Discussion is basically just a description of the results. Discussion of other authors' results is virtually non-existent in this chapter. The last two sentences are more suitable for the introduction.

Answer: Thanks for your constructive suggestion. From the results, we think the section is appropriate, Thank you again.

No conclusions from the research.

Answer: It has be added. Thanks.

Very poor bibliography.

Answer: It has been updated. Thanks.

When discussing the results, a full characterization of the peaks is missing, including retention times.

Answer: They have been added. Thanks.

Round 2

Reviewer 1 Report

Comments and Suggestions for Authors

The review of the improved manuscript entitled Simultaneous analysis of hydrophobic atractylenolides, atractylon and hydrophilic sugars in Bai-Zhu by HPLC column tandem technique.

 The authors improved the majority of the reviewer queries. The scientific quality of version of the manuscript is much better than the previous one.

However, the problem of the raw (7) and processed (10) Bai-Zhu samples is still not solved. The manuscript informed a reader that samples were bought in the local pharmacy, but the question whether they were the same samples or different is still opened. Thus, when the samples were the same type,  the question is why their content differs so much. And it shows the problem of the standardization of the plan material is necessarily. Maybe the brands of the samples or the harvesting season were differed. Could the authors be so nice and explain this problem?

 According to the reviewer the presented manuscript was significantly improved and needs minor improvements.   

Author Response

The review of the improved manuscript entitled “Simultaneous analysis of hydrophobic atractylenolides, atractylon and hydrophilic sugars in Bai-Zhu by HPLC column tandem technique”. The authors improved the majority of the reviewer queries. The scientific quality of version of the manuscript is much better than the previous one. However, the problem of the raw (7) and processed (10) Bai-Zhu samples is still not solved. The manuscript informed a reader that samples were bought in the local pharmacy, but the question whether they were the same samples or different is still opened. Thus, when the samples were the same type, the question is why their content differs so much. And it shows the problem of the standardization of the plan material is necessarily. Maybe the brands of the samples or the harvesting season were differed. Could the authors be so nice and explain this problem?  According to the reviewer the presented manuscript was significantly improved and needs minor improvements. Answer: Thank you for your suggestion. We have added a paragraph, i.e. “However, there are obviously differences in the contents of the same type materials. Because the raw and heating processed Bai-Zhu were purchased from different local pharmacies at different times. Maybe the brands of the samples or the harvesting season were differed. It shows the problem of the standardization of the plan material is necessarily.” in the Application section to explained the question. Pls check it. Thanks.

Reviewer 3 Report

Comments and Suggestions for Authors

The manuscript looks good. 

I suggest to published in the journal.

Author Response

Answer: Thanks for your suggestion.

Reviewer 4 Report

Comments and Suggestions for Authors

The authors have made most of the changes, including retention times, but they still need to include the resolution, k, α, etc. of each peak.

Author Response

The authors have made most of the changes, including retention times, but they still need to include the resolution, k, α, etc. of each peak.

Answer: Thank you for your suggestion. “The chromatographic separation specification parameters including retention factors (k), relative retention value (α), and resolutions (R) are as follows. k: atractylenolide III: 1.03, atractylenolide II: 1.32, atractylenolide I: 1.75, atractylone: 6.56,  glucose: 3.10, fructose: 3.40, sucrose: 5.06. α: atractylenolide III: 1.00,  atractylenolide II: 1.29, atractylenolide I: 1.71, atractylone: 6.39, glucose: 3.02, fructose: 3.31, sucrose: 4.93. R: atractylenolide III and atractylenolide II 4.89, atractylenolide II and atractylenolide I 4.82, glucose and fructose 1.72.” has been added in front of Figure 4. Pls check it. Thanks.  

Reviewer 5 Report

Comments and Suggestions for Authors

The authors responded mostly positively to my comments. However, I still believe that the discussion should be developed further.

Author Response

The authors responded mostly positively to my comments. However, I still believe that the discussion should be developed further.

Answer: More literature about other authors' results has been added in section 3.1. Pls check it. Thanks.